# MoS_2_-Based Memristor: Robust Resistive Switching Behavior and Reliable Biological Synapse Emulation

**DOI:** 10.3390/nano13243117

**Published:** 2023-12-11

**Authors:** Yongfa Ling, Jiexin Li, Tao Luo, Ying Lin, Guangxin Zhang, Meihua Shou, Qing Liao

**Affiliations:** 1Guangxi Key Laboratory of Precision Navigation Technology and Application, Guilin University of Electronic Technology, Guilin 541004, China; yfling73@163.com (Y.L.); 21022303073@mails.guet.edu.cn (J.L.); 21022303115@mails.guet.edu.cn (T.L.); 22022303085@mails.guet.edu.cn (Y.L.); 14795402135@163.com (G.Z.); 2School of Mechanical and Electronic Engineering, Hezhou University, Hezhou 542899, China

**Keywords:** MoS_2_, memristor, two-dimensional material, resistive memory, synapse, synaptic behavior

## Abstract

Memristors are recognized as crucial devices for future nonvolatile memory and artificial intelligence. Due to their typical neuron-synapse-like metal–insulator–metal(MIM) sandwich structure, they are widely used to simulate biological synapses and have great potential in advancing biological synapse simulation. However, the high switch voltage and inferior stability of the memristor restrict the broader application to the emulation of the biological synapse. In this study, we report a vertically structured memristor based on few-layer MoS${}_{2}$. The device shows a lower switching voltage below 0.6 V, with a high ON/OFF current ratio of $10^{4}$, good stability of more than 180 cycles, and a long retention time exceeding 3 × 10${}^{3}$ s. In addition, the device has successfully simulated various biological synaptic functions, including potential/depression propagation, paired-pulse facilitation (PPF), and long-term potentiation/long-term depression (LTP/LTD) modulation. These results have significant implications for the design of a two-dimensional transition-metal dichalcogenides composite material memristor that aim to mimic biological synapses, representing promising avenues for the development of advanced neuromorphic computing systems.

## 1. Introduction

With the growing need to process and store vast amounts of data, there is an urgent demand to upgrade the computing hardware from complementary metal oxide semiconductor (CMOS) electronics. This is because CMOS electronics face limitations such as high energy consumption, heat dissipation issues, and performance bottlenecks when dealing with the challenges posed by large-scale data. Researchers are actively exploring new technologies and materials to pioneer advancements in computing hardware and meet the growing demand [1]. The memristor proposed by Chua et al. has nonvolatile resistive switching behavior and hysteretic I-V characteristics and is considered to be one of the most promising candidates for future nonvolatile devices and next-generation AI memory devices [2,3]. Traditional memristors based on transition metal oxides and organic molecules require high energy consumption, relatively complex manufacturing processes, and relatively slow operation. These factors limit their widespread application and further development [4]. Conversely, memristors based on two-dimensional transition-metal dichalcogenides (TMD) materials shows advantages such as nanoscale thickness, highly controllable resistive characteristics, low power consumption, and high speed. This makes them ideal for memristor research and applications [5,6,7]. By utilizing 2D TMDs materials with an ultra-thin layered structure as the resistive switching layer material, it overcomes the limitations of traditional resistive switching layer materials in reducing the size of memristors, efficiently enhances the responsiveness to external stimuli, and minimizes the device dimensions as close to the biological synapse as possible [8,9,10].

Based on these advantages of 2D transition metal chalcogenides, several groups have achieved interesting performance milestones, such as the main functions of MoS${}_{2}$ in biological synaptic behavior, including short-term plasticity (STP), spike-timing dependent plasticity (STDP), paired-pulse facilitation (PPF), and long-term potentiation (LTP)/depression (LTD) [11,12,13,14,15]. For instance, Renjing Xu et al. reported a vertical memristor based on the Cu/MoS${}_{2}$/Au structure, where these atomically thin 2D materials facilitate the achievement of low switching voltages and enable Long-Term Potentiation (LTP) and Spike-Timing-Dependent Plasticity (STDP), but this low switching voltage is achieved at the expense of device endurance (~20 cycles) and switching ratio (<10) [16]. Additionally, Zhao et al. reported a vertical heterostructure memristor based on MoS${}_{2}$/WO${}_{3}$, where this memristor successfully simulated biological synaptic behaviors (such as PPF and EPSC) and demonstrated the potentiation/depression of synaptic behaviors under different bias voltages [17]. Recently, Eunho Lee et al. reported an Au/MoS${}_{2}$/Au memristor device, with a switch voltage of over 2V, that simulates some artificial synapse behaviors [18]. It can be observed that memristors show immense potential in emulating biological synapses, but face challenges in the high switch voltage and inferior stability. Most of the MoS${}_{2}$-based memristors reported so far are predominantly fabricated by exposing MoO${}_{3}$ or Mo to a sulfur-containing atmosphere, inducing a sulfidation reaction to form MoS${}_{2}$ films. However, the film quality produced by this growth method is suboptimal, potentially containing defects and impurities. These defects and impurities have the potential to influence the electronic transport properties of MoS${}_{2}$, leading to electron scattering phenomena, thereby affecting the stability and switch ratio of the devices.

Here we report on a Molten-salt-assisted CVD method to synthesize large-size MoS${}_{2}$ on SiO${}_{2}$/Si substrates and further explore vertical 2D material memristors with atomic-scale thickness based on an Ag/MoS${}_{2}$/Pt cell. The device exhibited low SET/RESET voltages (<±0.6 V), a large switching on/off ratio up to $10^{4}$, a clearly extended endurance up to 180 switching cycles, and presents a long retention time for both HRS and LRS over 3 × $10^{3}$ s. We further explore and confirm that the resistance switching mechanism of the memristor was caused by the formation and dissolution of the metal conducting filaments. Additionally, the device successfully simulates various biological synaptic functions, such as alterations in potentiation/depression conductance, paired-pulse facilitation (PPF), long-term potentiation (LTP), and long-term depression (LTD). It is expected to effectively improve the stability, reduce the switching voltage of memristors based on ultra-thin MoS${}_{2}$, and lay the foundation for the realization of low-power neuromorphic computers in the future.

## 2. Materials and Methods

### 2.1. CVD Growth of MoS${}_{2}$

In this study, the resistive layer material MoS${}_{2}$ was obtained via a molten-salt-assisted chemical vapor deposition (CVD) method [19]. As shown in Figure 1a, a dual-temperature tube furnace, including a low-temperature heating zone (upstream of the quartz tube) and a high-temperature heating zone (downstream of the quartz tub), was used for the experiment. In the low-temperature heating zone, a quartz boat containing sulfur powder (20 mg, 99.98% purity, Sigma-Aldrich, St. Louis, MO, USA) was placed, and in the high-temperature heating zone, a quartz boat containing MoO${}_{3}$ (4 mg, 99.97% purity, Alfa Aesar, Haverhill, MA, USA) and NaCl (0.5 mg, 99.98% purity, Sigma-Aldrich) was placed. The quartz boats in the two heating zones were 29 cm apart. A silicon wafer with a 300 nm oxide layer was selected as the substrate. Before starting the growth, the growth substrate (SiO${}_{2}$/Si) was rinsed in isopropanol, acetone, ethanol, and deionized water for 10 min and then dried with nitrogen (N${}_{2}$). Afterwards, the cleaned silicon substrate was placed face up on a quartz boat equipped with an MoO${}_{3}$ and NaCl source. The tube furnace growth system was pumped down to ~0.1 Pa, then the quartz tube was filled with argon (Ar) and pumped down to ~0.1 Pa again; this process was repeated three times to minimize the presence of air and impurities in the quartz tube. For CVD growth, a constant pressure of 200 sccm (standard cubic centimeters per minute) was noted as the argon (Ar) flow rate. The low-temperature and high-temperature heating zones increased the temperature to 120 °C and 800 °C, respectively, in 20 min. The low-temperature zone was then heated at 8 °C/min for 10 min to 200 °C, and the high-temperature zone was maintained at 800 °C for 10 min. After growth was complete, the system was naturally cooled to room temperature.

### 2.2. Transfer of MoS${}_{2}$ and Device Fabrication

The device structure of the MIM-type memristor is shown in Figure 1b. The fabrication process of the memristors is as follows. First, the bottom electrode (Pt 100 nm/Ti 80 nm) was obtained via magnetron sputtering. Next, we transferred the CVD-grown MoS${}_{2}$ to a substrate with pre-fabricated bottom electrodes (Pt/Ti/SiO${}_{2}$/S): MoS${}_{2}$ films were coated with a layer of polymethylmethacrylate (PMMA), where the coating was applied via rotational coating. (Stage 1: 10 s of rotation at 500 rpm; Stage 2: 60 s of rotation at 3000 rpm). After completing the coating, the samples were placed in an oven at 100 °C for 10 min to cure the PMMA and form a protective layer with the MoS${}_{2}$ film. Next, the PMMA-encapsulated MoS${}_{2}$ film was placed into a NaOH(1M) solution, and the SiO${}_{2}$ was etched to obtain PMMA/MoS${}_{2}$ by heating at 100 °C for 30 min. After completing the NaOH treatment, the PMMA/MoS${}_{2}$ film was transferred to deionized water (DI water) to remove the etchant and residues. Then, the PMMA-covered MoS${}_{2}$ film was fished using a clean Pt/Ti/SiO${}_{2}$/Si substrate and baked on a hot plate at 100 °C for 10 min. This was followed by the removal of PMMA in acetone. Finally, a 300 nm thick, 100 μm diameter silver-top electrode (TE) was deposited on MoS${}_{2}$/Pt using a thermal evaporation technique and a shadow mask.

## 3. Results and Discussion

Successful preparation of the few-layer MoS${}_{2}$ is a key factor in achieving device performance. The crystal quality of MoS${}_{2}$ crystals synthesized via the traditional chemical vapor deposition (CVD) method without molten-salt-assisted growth is not ideal, and the prepared devices have poor stability and a relatively low on/off ratio [16]. To enhance the stability of memristors based on few-layer MoS${}_{2}$, we employed molten-salt-assisted chemical vapor deposition to grow MoS${}_{2}$ films. To assess the quality and thickness of our grown films, we conducted SEM (Jeol JSM-IT500HR , Tokyo, Japan) and Raman (HORIBA LabRAM HR Evolution, 532 nm) characterizations on the samples. Figure 1c shows the SEM image of a MoS${}_{2}$ film grown on a SiO${}_{2}$ substrate. In the SEM images, two distinct regions can be observed: a bright white area representing the island, and a flat dark area. To investigate the differences between these two regions, Raman tests were performed.

Two high-intensity peaks were detected in the Raman spectrum of the bright white area, at 409.9 cm${}^{-1}$ and 385.5 cm${}^{-1}$, corresponding to the Raman characteristic peaks *A*${}_{1g}$ and *E*${}_{2g}^{1}$ of MoS${}_{2}$, respectively. By calculating the peak difference $\Delta =A_{1g}-E_{2g}^{1}$ = 24.4 cm${}^{-1}$, we determined that the number of MoS${}_{2}$ layers in this region is more than four. Similarly, two characteristic peaks were also detected in the Raman spectrum of the dark area, at 409.4 cm${}^{-1}$ and 386.0 cm${}^{-1}$, respectively. Calculating the peak difference $\Delta =A_{1g}-E_{2g}^{1}$ = 23.4 cm${}^{-1}$, it indicated that the number of MoS${}_{2}$ layers in the dark area is three. This result indicates that we have achieved the growth of continuous full-covered MoS${}_{2}$ films. We performed Raman characterization of the as-grown MoS${}_{2}$ flakes and the exact flakes after transfer to study the quality of the as-growth and transferred MoS${}_{2}$ films (Figure 1d). In Figure 1d, the *E*${}_{2g}^{1}$ and *A*${}_{1g}$ modes of the Raman band of as-growth MoS${}_{2}$ are located at 386.5 cm${}^{-1}$ and 410.4 cm${}^{-1}$, respectively. The measured frequency difference ($\Delta \;=\;A_{1g}\;-\;E_{2g}^{1}$) is 23.9 cm${}^{-1}$. About the transferred MoS${}_{2}$ flakes, the position of the *E*${}_{2g}^{1}$ mode remains unchanged, while the *A*${}_{1g}$ mode shifts to a higher frequency near 409.47 cm${}^{-1}$. The FWHM of *E*${}_{2g}^{1}$ before and after transfer are 5.1 cm${}^{-1}$ and 5.2 cm${}^{-1}$, respectively, and the FWHM of *A*${}_{1g}$ are 7.8 cm${}^{-1}$ and 7.7 cm${}^{-1}$, respectively. These results indicate that the MoS${}_{2}$ wafer layer exhibit good uniformity and that the transfer process did not impair its quality.

The current-voltage (I–V) characteristics of the Ag/MoS${}_{2}$/Pt memristor were investigated to study their resistive switching behavior. We used a semiconductor analyzer (Keithley 2636B) hooked with a probe station (Figure 2a) to measure the I–V characteristic curve. In the course of electrical measurements, positive and negative biases were determined based on the current flow between the top and bottom electrodes. The prepared devices were tested electrically at ambient conditions at room temperature. During the tests, the BE electrode was grounded and a voltage sweep was applied to the Top electrode, with a compliance current of 0.01 A employed to prevent device breakdown. The I–V characteristic curve of the Ag/MoS${}_{2}$/Pt device is presented in Figure 2b. The applied voltage to the device cycled from 0 V → 0.6 V→ 0 V → −1.5 V → 0 V. As shown in Figure 2b, before voltage application begins, the device is in its original state and exhibits very high resistance (approximately 1 MΩ). When the applied voltage increases from 0 V to 0.4 V, the device exhibits a gradual increase in current with the increase in voltage. The voltage further increases from 0.4 V to 0.6 V and the current rises sharply from 10${}^{-6}$A to 10${}^{-2}$A and mutates at V = 0.42($V_{\mathrm{set}}$), indicating that the device switches from HRS to LRS (switching from MΩ to Ω). Reverse bias is applied to the device, wherein the voltage is swept from 0.6V to 0V to −1.5V. Within the voltage range of 0.6V to −0.56V, the device maintains the low-resistance state (LRS). As the scan voltage reaches V = −0.56, $V_{\mathrm{reset}}$, the current undergoes a sharp drop from $10^{-2}\;A$ to $10^{-5}\;A$, indicating the transition of the device from LRS to a high-resistance state (HRS). The states “1” and “0” correspond to the ON and OFF states, respectively, with the former indicating a higher current state and the latter representing a lower current state. The transformation from the OFF state to the ON state corresponds to the programming or writing process in the digital memory cell. Once the conversion is complete, the device remains on even when the power supply is removed, as evidenced by the sustained high current flow in the upper part of Figure 2b. Applying a reverse bias voltage of approximately −0.56 V (referred to as the erase voltage) achieves the transformation from the ON state to the OFF state. When the device current decreases to five orders of magnitude, the device returns to its initial OFF state, equivalent to the erasure process in digital memory. From the above I-V test results, we are demonstrating the device’s electrical hysteresis behavior that exhibits a current ratio exceeding $10^{4}$.

It is crucial to address the issue of instability during set/reset cycling, which is a critical concern for 2D TMD material memory resistors [7,20]. Typically, it is challenging to achieve more than 100 reliable switching cycles for memristors composed of monolayer or few-layer ultrathin 2D materials (less than 10 nm in thickness). In order to prove the resistance switching stability of the device, we tested the resistance switching characteristics of the device in pulse mode, as shown in Figure 2c, where the Ag/MoS${}_{2}$/Pt device can reach 180 switching cycles in pulse mode, and the device resistance in HRS was approximately 1 MΩ, and in LRS, it was dozens of ohms, aligning with the resistance state during the I–V characteristic test. Data retention is a significant performance indicator for storage devices, reflecting their data retention capability. As illustrated in Figure 2d, the memristor maintained a duration of 3 × 10${}^{3}$ s in both LRS and HRS. In addition, we prepared a device with the same structure using MoS${}_{2}$ prepared via chemical vapor deposition without molten salt assistance. Performing I–V characterization on the device, the test results are shown in Figure 3a. Based on the test results, it can be observed that the stability of the device is relatively poor when grown without molten salt assistance. After approximately 20 cycles, the device exhibits only subtle resistive switching characteristics, and the maximum switching ratio of the device is merely 10${}^{2}$. We tested the I-V characteristics of MoS${}_{2}$ using molten-salt-assisted growth for 30 cycles, as shown in Figure 3b. The device showed the stable device’s electrical hysteresis behavior within 30 cycles. Compared with devices without molten-salt-assisted growth, it shows a higher switching ratio and a more stable resistance switching phenomenon.

Furthermore, this device exhibits free-forming bipolar resistance switching behavior, implying that the formation and dissolution of metal conductive filaments is the dominant mechanism of memristor resistance switching [21,22]. To further investigate the transmission mechanism, the I-V characteristic curve of the memristor was subjected to piecewise linear fitting and analyzed in a log–log scale, as shown in Figure 3c. During the SET operation, the conductance mechanism of the device adhered to Ohm’s law and was fitted with a slope of approximately 0.97. In the low-voltage region (region A), the I–V characteristic curve exhibits a linear relationship (I∝V), which suggests that the conductance of the device was governed by the thermal excitation of a finite number of minority carriers present in the high-resistive state (HRS). As the voltage is gradually increased, the slope of the I–V characteristic curve rises to about 2.14 (region B), showing the nonlinear behavior of the Child conduction mechanism($I\propto V^{2}$). At this phase of the process, the flow of the current was fully regulated by the space charge, which limits the additional introduction of carriers into the resistive layer. Once the voltage is increased to V${}_{\mathrm{set}}$, all of the traps are filled, leading to a rapid increase in the large number of free carriers and a sharp increase in the current, which changes the device from the HRS state to the LRS state. According to the above analysis, the electrical behavior of the device can be attributed to the space charge limiting conduction (SCLC) mechanism in the filled traps [23]. In the LRS state, the slope of the curve is about 1.04 (region C), indicating that the conductive behavior of the device is in accordance with the ohmic conduction mechanism, implying the formation of metal conducting filaments (CFs) inside the device. The different conductive characteristics of the Ag/MoS${}_{2}$/Pt device in the HRS and LRS, indicating that the resistive switching behavior of the device is controlled by the metal conducting filaments [24,25]. The presence of this on-state mechanism suggests that the metal conducting filaments in the MoS${}_{2}$ layer were crucial in regulating the conductivity characteristics of the devices.

Based on the above discussion, it is indicated that the operational model of the device relies on the formation and dissolution of conductive filaments. We propose a non-volatile resistive switching model to analyze the operational mechanism of the Ag/MoS${}_{2}$/Pt device, as illustrated in Figure 4. Figure 4a depicts the initial state of the device without applied voltage. When a positive voltage is applied to the active metal electrode, metal atoms on the active electrode undergo oxidation, transforming into metal ions. These ions are injected into the MoS${}_{2}$ dielectric layer and subsequently reduced to metal atoms at the inert electrode, as shown in Figure 4b. With the accumulation of reduced ions, a conductive metal filament forms between the inert and active electrodes, illustrated in Figure 4c. This filament significantly reduces the device resistance, completing a the “write” process. The “erasure” process is analogous to “write” process. When a negative voltage is applied to the active electrode, metal ions depart from the filament (Figure 4d), returning to the active electrode. This causes the conductive filament to dissolve, returning the device to HRS. To provide additional support, we studied an MoS${}_{2}$-based memristor using the Ag (active metal) and Au (inert metal) as TE. The I–V characteristics of the two devices were tested separately, and the test results are shown in Figure 3b and Figure 3d. In Figure 3b, when a sweep voltage applied to the device cycled from 0 V → 0.6 V → 0 V → −1.5 V → 0 V, the memristor with the active electrode (Ag) switches from HRS to LRS, exhibiting electrical hysteresis behavior. When a positive voltage is applied to the Ag/MoS_2_/Pt device, the active Ag electrode undergoes ionization at the interface, leading to the injection of cations into the dielectric layer (MoS${}_{2}$ film). These cations create conductive pathways for electrons, resulting in a low-resistance state (LRS) corresponding to the process from Figure 4b to Figure 4c. Simultaneously, reversible redox reactions ensure that the device can return to a high-resistance state (HRS) during negative voltage scanning, corresponding to the process from Figure 4c to Figure 4d. However, when a sweep voltage applied to the Au/MoS${}_{2}$/Pt device cycled from 0 V → 15 V→ 0 V → −15 V → 0 V, no electrical hysteresis behavior was observed for the device, as shown in Figure 3d. According to the previous literature [18], when sandwiching the MoS${}_{2}$ dielectric layer between two inert electrodes, achieving the electrical hysteresis behavior in devices with this structure relies on the presence of vacancies or defects within the dielectric layer. The results of two different experiments indicate that the vacancies or defects naturally occurring in molybdenum disulfide growth assisted by molten salt CVD were inadequate in triggering RS behavior. The active metals can be utilized to enhance the electroplating process and assist in the injection of positive ions, leading to the observed resistive switching behavior.

According to the results of our experiments, we found that our device not only exhibits bipolar characteristics, but also possesses some of the characteristics of an analog biological synapse. The device exhibited gradual adjustments in positive and negative currents with changes in the voltage, which serves as a crucial foundation for bidirectional resistance modulation in biological synapse simulators. This may be due to the need for bi-directional progressive regulation of amnestic conductance when modeling synaptic potentiation and depression processes and plastic synapses [26]. Therefore, it is suitable for use as an artificial synapse. We performed a direct current (DC) voltage scan of the device and measured the gradual modification of the conductance in DC mode to evaluate the potentiation and depression effects of the memristor. As shown in Figure 5a, when five positive voltages were applied, the conduction current increases gradually with scanning cycles and always increased based on the previous scan. Subsequently, the conduction current decreases with scanning cycles over multiple scans using a 0 V to −0.5 V negative polarity voltage as shown in Figure 5b. Additionally, the starting point of the current for each subsequent voltage sweep is always the end point of the last sweep current. The plot of the relationship with time is given in Figure 5c. The above test results exhibit behaviors that are comparable to the phenomena of synaptic potentiation and inhibition, which are essential features required for the development of devices capable of simulating biological synapses.

Figure 6a illustrates the structure of a synapse in a neural network, comprising a presynaptic dendrite, a postsynaptic dendrite, a synaptic cleft, and neurotransmitters. When the presynaptic terminal releases neurotransmitters to the receptor in the postsynaptic dendrite, it induces a change in the postsynaptic current. This mimicry of biological synapses is expected to have important applications in artificial intelligence and neuroscience. We can observe potentiation and depression effects via pulse mode experiments (see Figure 6b). For example, we applied a sequence of 100 consecutive positive pulses (with a pulse amplitude of 0.5 V and a pulse duration of 0.5 s), followed by a sequence of 100 consecutive negative pulses (with a pulse amplitude of −0.5 V and a pulse duration of 0.5 s), with a time interval between pulses of about 0.5 s. After each pulse, we measured the resistance of the device and read a voltage of 10 mv. As anticipated from prior DC scanning experiments, positive pulses led to a gradual decrease in the device’s resistance (potentiation), while negative pulses resulted in a gradual increase in the device’s resistance (depression). By applying the initial 100 positive pulses, we observed a gradual increase in the current, indicating long-term potentiation (LTP). Subsequently, through the application of 100 negative pulses, the current gradually decreased, implying that the device exhibits long-term depression (LTD) functionality. Therefore, memristors based on molybdenum disulfide hold promising applications in synaptic simulation and neural network computation.

PPF is a form of short-term plasticity, when a neuron obtains two consecutive stimuli, the postsynaptic response of the second pulse is greater than the first one [27]. In Figure 6c, I${}_{1}$ represents the increase in current during the first pulse, while I${}_{2}$ represents the increase in current during the second pulse. To gain a better understanding of how PPF behaves in your device, Figure 6d plots the changes in the PPF index as the time interval (Δ*t*) between the two pulses varies. From Figure 6d, it can be observed that as the pulse interval decreases, the *PPF* exponent increases, providing further evidence of the reliability of the device in the application to biological synaptic neural networks. *PPF* serves as an indicator of synaptic plasticity, which can be determined using the following function:(1)$$PPF=\frac{I_{2}-I_{1}}{I_{1}}\times 100\%$$

This plot allows you to examine how the *PPF* index is influenced by the timing of the successive stimuli in your specific device. From the curve of the *PPF* index-(Δ*t*), the following equation enables the extraction of the characteristic parameters of our device [28]:(2)$$PPF=C_{1}\times \exp\left(-\Delta t/\tau_{1}\right)+C_{2}\times \exp\left(-\Delta t/\tau_{2}\right)$$
where −Δ*t* denotes the inter-pulse interval, *C*${}_{1}$ and *C*${}_{2}$ denote the initial facilitation magnitudes, and $\tau_{1}$ and $\tau_{2}$ are two fitting constants corresponding to the fast decaying terms and slow decaying terms, respectively. The estimated values of $\tau_{1}$ and $\tau_{2}$ were found to be 57.55 ms and 609.22 ms, respectively, according to the results obtained. The measurement results show that the device based on MoS${}_{2}$ behaves similarly to a synapse, indicating its potential as a candidate for functionally simulating biological synapses.

## 4. Conclusions

In summary, we propose a molten-salt-assisted CVD method to synthesize large-size MoS${}_{2}$ on SiO${}_{2}$/Si substrates and further explore vertical 2D material memristors with atomic-scale thicknesses based on an Ag/MoS${}_{2}$/Pt cell, demonstrating a remarkable switching ratio of $10^{4}$, low Set/Reset voltages (0.42 V/−0.56 V), excellent endurance of over 180 set/reset cycling processes, and a long retention time exceeding 3 × 10${}^{3}$ s. Compared to the other few-layer MoS${}_{2}$ memristor-based literature, our device exhibits stable cycling characteristics and lower switching voltages. Furthermore, we further explore and confirm that the RS mechanism is mainly associated with the formation and dissolution of metallic conductive filaments. In addition to exhibiting robust resistive switching behavior, our study successfully demonstrates certain synaptic functions, such as paired-pulse facilitation (PPF) and long-term potentiation/long-term depression (LTP/LTD), which are pertinent to biological functions. These findings clearly indicate that the MoS${}_{2}$ film-based memristor holds remarkable potential for emulating biological synapses, providing an avenue for the construction of an artificial neural network.

## Figures and Tables

**Figure 1 nanomaterials-13-03117-f001:**
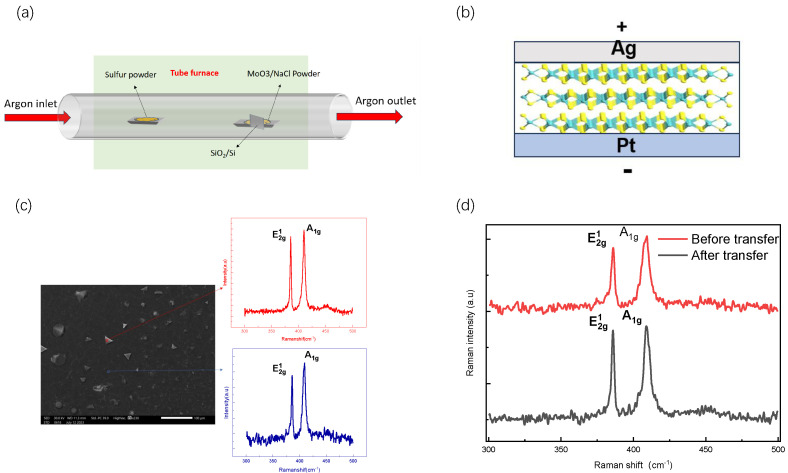
(**a**) The modified CVD system for MoS${}_{2}$ growth. (**b**) Schematic diagram of Ag/MoS${}_{2}$/Pt device structure. (**c**) SEM images of the MoS${}_{2}$. (**d**) As-growth and transferred Raman spectra of MoS${}_{2}$.

**Figure 2 nanomaterials-13-03117-f002:**
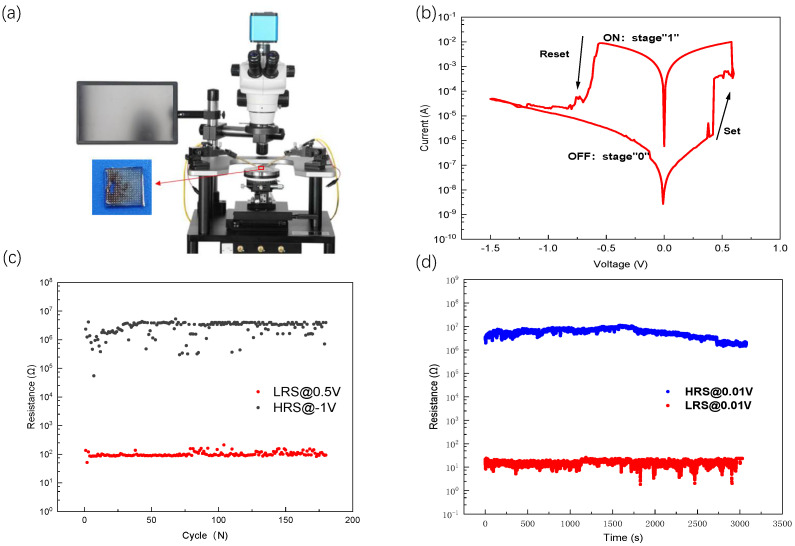
Electrical characterization of the Ag/MoS${}_{2}$/Pt memristor. (**a**) Probe station for testing I–V characteristics, the picture inserted in the lower-left corner is the device physical picture of the device Ag/MoS${}_{2}$/Pt. (**b**) Semilog plot of representative I-V curves for bipolar resistive switching in the Ag/MoS${}_{2}$/Pt memristor. (**c**) Endurance properties of the device. (**d**) The resistive-state retention time on LRS and HRS states. The read voltage during the test was set to 10 mv.

**Figure 3 nanomaterials-13-03117-f003:**
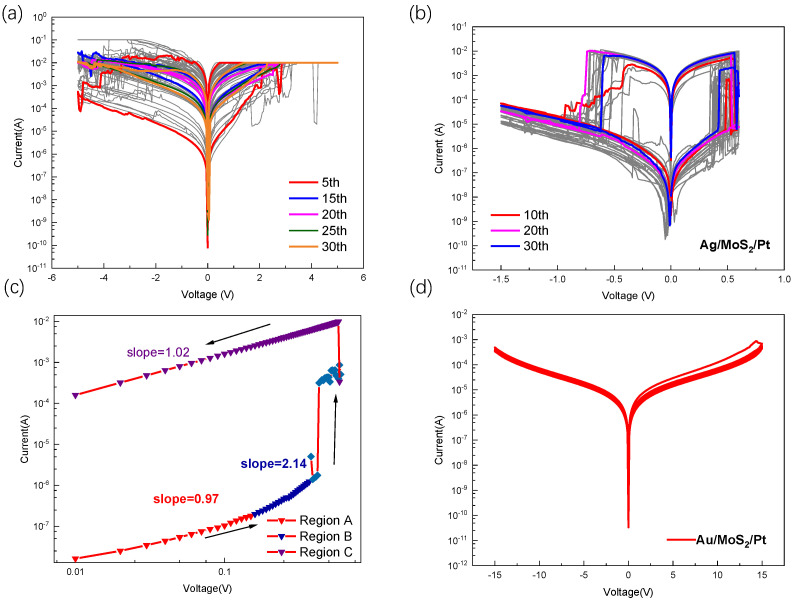
(**a**) I–V curves of Ag/MoS${}_{2}$/Pt without molten-salt-assisted growth. (**b**) I–V curves of Ag/MoS${}_{2}/$Pt using molten-salt-assisted growth. (**c**) Log–log plot of Ag/MoS${}_{2}$/Pt device’s I–V characteristics with a linear fitting curve under forward bias. (**d**) I–V curves of Au/MoS${}_{2}$/Pt memristor.

**Figure 4 nanomaterials-13-03117-f004:**
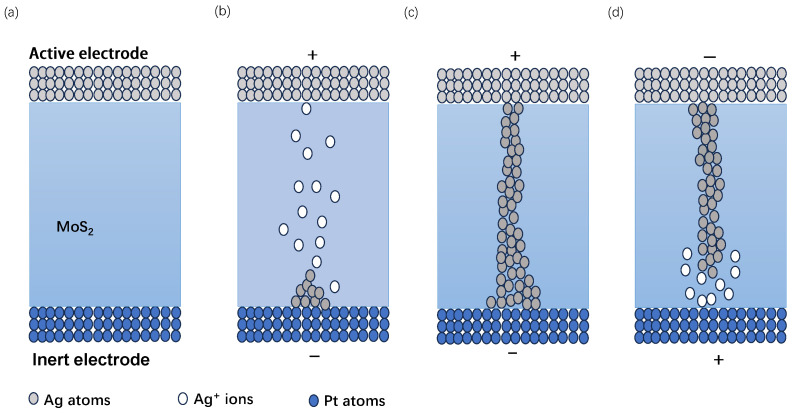
Schematic illustration of CFs’ formation and dissolution processes in Ag/MoS${}_{2}$/Pt. (**a**) The initial state of the device before voltage is applied. (**b**) Ag${}^{+}$ cation injects into MoS${}_{2}$ layer, Ag${}^{+}$ cations are implanted into the MoS${}_{2}$ layer, and then a reduction reaction occurs at the inert electrode to form Ag atoms. (**c**) CFs are formed between two electrodes. (**d**) The CFs dissolve when a reverse voltage is applied.

**Figure 5 nanomaterials-13-03117-f005:**
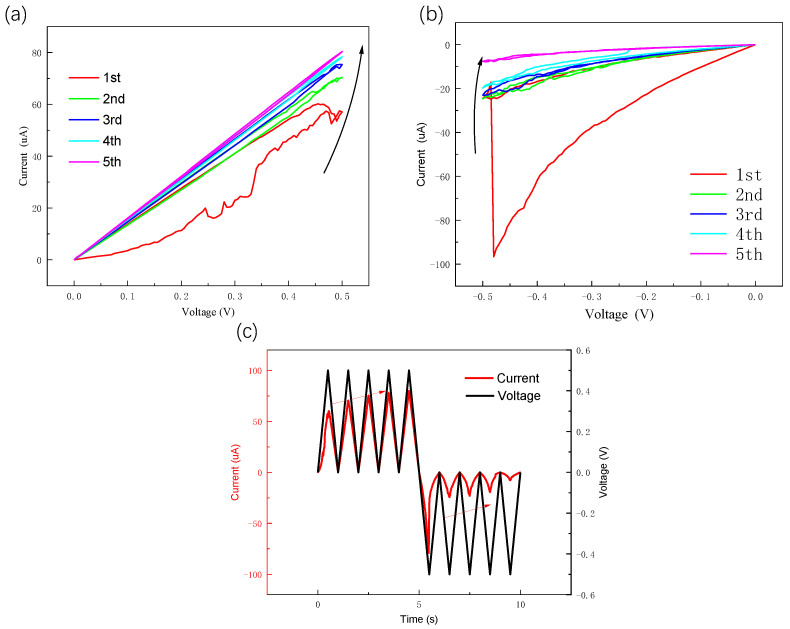
Synaptic plasticity of Ag/ MoS${}_{2}$/Pt memristors operated in DC sweep mode. (**a**) Forward DC scan, potentiation response of Ag/MoS${}_{2}$/Pt device with repeated voltage sweeps. (**b**) Reverse DC scan, depression response of Ag/MoS${}_{2}$/Pt device with repeated voltage sweeps. (**c**) Variation in voltage and current over time.

**Figure 6 nanomaterials-13-03117-f006:**
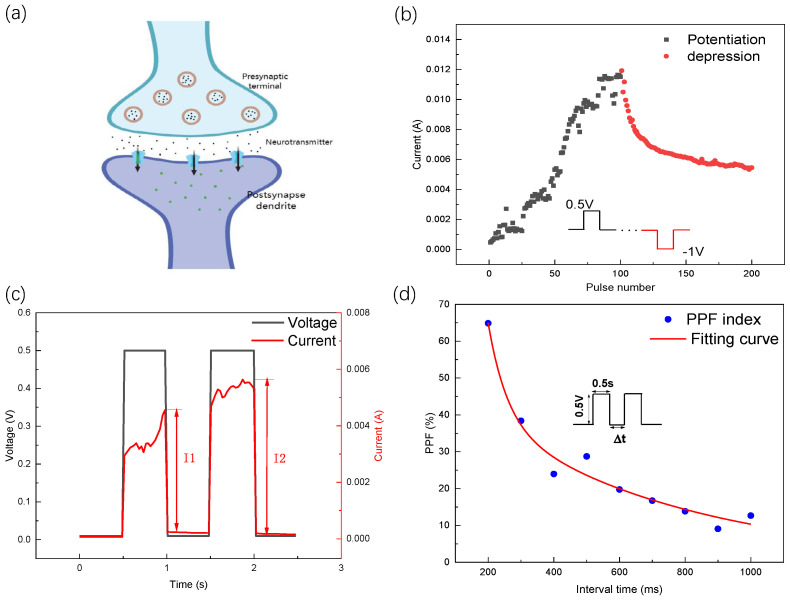
Synaptic measurements of the proposed MoS${}_{2}$ memristor. (**a**) Structure of synapse in neural network. (**b**) Potentiation and depression under consecutive positive and negative pulses (pulses are used at 0.5 V/−1 V. The pulses’ width and interval time were all 0.5 s). (**c**) Paired pulse facilitation curve. (**d**) PPF ratio with the different pulses’ interval time; the red line in the figure is the fitting curve.

## Data Availability

Upon making a reasonable request, the corresponding author will provide access to the data that substantiates the results of this study.

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
