# Peer review of "MoS2-Based Memristor: Robust Resistive Switching Behavior and Reliable Biological Synapse Emulation"

_nanomaterials, 2023, doi:10.3390/nano13243117_

Round 1
Reviewer 1 Report
Comments and Suggestions for Authors
This manuscript presented a vertical structure memristor based on Ag/MoS2/Pt cell. Although authors presented various electrical characterizations and representative plots of memristor, there is lack of novelty and progressiveness of the thesis in this research. Those properties that was presented in this research as novelty are already shown in previous research and there is no originality. The proposed experimental data does not contain major reaction which is newly discovered in this research and the detailed reaction mechanism doesn’t exist. In addition, the quality of those data is not enough for the publication as it is. Although authors suggested the reliability of the platform as biological synapse emulation, the experimental data is not enough for supporting the opinion.
As a result, I can’t recommend this manuscript for publication in Nanomaterials
Comments on the Quality of English LanguageModerate editing of English language required.
Reviewer 2 Report
Comments and Suggestions for Authors
The authors discussed the following topics: Robust resistive switching behavior and reliable biological synapse emulation in MoS2-based memristors. This paper discusses a vertical 5 memristor structure based on few layers of MoS2. The device shows a lower switching voltage below 6 0.6V, a high ON/OFF current ratio of 104, good stability of more than 180 cycles, and a long 7 retention time exceeding 3×103 s. Additionally, the device simulates potential/depression propagation, paired-pulse facilitation (PPF), and long-term potentiation/long-term depression (LTP/LTD). Before I publish, I have a few minor concerns
1. The figure quality is poor, even nothing is clearly visible.
2. The author should provide more details about the experimental measurement part with a figure of the setup.
3. The author needs to explain the working principle of the idea, which is not clearly explained.
Comments on the Quality of English LanguageNo comments
Reviewer 3 Report
Comments and Suggestions for Authors
The paper presents a Ag/MoS2/Pt structure that can be used as a memristor. First, the fabrication process is described. Then, the memristive behavior is demonstrated. Finally, the authors show that their device has some of the characterisitics of a biological synapse. This is a regular paper, in my opinion. I have the following comments:
1) The figures are too small.
2) Regarding Figure 2 (b). There are nearly up to 2 orders of magnitude between cycles for the resistive values of the HRS. This is much more than for the LRS. Any ideas that could explain this ?
3) The legends of regions B and C are the same in Fig. 2 (d). Some arrows could be added to show the direction of the voltage sweep.
4) About Figure 3: the fact that the memristive behavior does not appear when the Ag is replaced with Au is very interesting. The authors should elaborate more. Could it be due to the workfunction difference of the electrode materials? Does the memristive behavior appear at a higher voltage ?
Comments on the Quality of English Language- "As show" should be "as shown", lines 70, 211 and 214
- line 95: "(PMMA). then" should be "(PMMA). Then"
- line 154: "we demonstrating" should be "we are demonstrating"
- line 202: "experiments found" should be "experiments, we found"
- line 203: "but also possesses analog biological synapse" should be "but also possesses some of the characteristics of an analog biological synapse", or something similar.
-In Figure 5 (c), "Volatage" in the axis and legend should be "Voltage"
Round 2
Reviewer 1 Report
Comments and Suggestions for Authors
Overall, the manuscript was revised and added additional explanation for supporting novelty, providing supportive experimental results. However, there are minor comments that authors should revise through the overall manuscript.
1. Those letters written on the data and graph are too small and indistinguishable in Figure 1, 2, 3, 4, 5, 6. Clarify the information for better readability.
2. In Figure 1b, schematic diagram of platform which was developed in this research show poor quality and not enough to contain the information and novelty asserting in overall manuscript.
3. Overall, the presentation of subscript and superscript is not segmented and needs revision.
As a result, I can recommend this manuscript for publication in Nanomaterials after minor revision.
Comments on the Quality of English Language
Overall, the presentation of subscript and superscript is not segmented and needs revision. Several typos and miswritten punctuation mark also must be revised for the publication in Nanomaterials.
